# Antimicrobial Resistance Profiles of *Escherichia coli* from Diarrheic Weaned Piglets after the Ban on Antibiotic Growth Promoters in Feed

**DOI:** 10.3390/antibiotics9110755

**Published:** 2020-10-29

**Authors:** Do Kyung-Hyo, Byun Jae-Won, Lee Wan-Kyu

**Affiliations:** 1Department of Veterinary Bacteriology and Infectious Diseases, College of Veterinary Medicine, Chungbuk National University, Cheongju 28644, Korea; pollic@chungbuk.ac.kr; 2Animal Disease Diagnostic Division, Animal and Plant Quarantine Agency, Gimcheon 39660, Korea; jaewon8911@korea.kr

**Keywords:** *Escherichia coli*, antimicrobial resistance, swine, weaned piglet, antibiotic growth promoters

## Abstract

This study aimed to survey the antimicrobial resistance profiles of 690 pathogenic *Escherichia coli* isolates obtained from Korean pigs with symptoms of enteric colibacillosis between 2007 and 2017, while assessing the change in antimicrobial resistance profiles before and after the ban on antibiotic growth promoters (AGPs). Following the Clinical and Laboratory Standards Institute guidelines, the antimicrobial resistance phenotype was analyzed through the disk diffusion method, and the genotype was analyzed by the polymerase chain reaction. After the ban on AGPs, resistance to gentamicin (from 68.8% to 39.0%), neomycin (from 84.9% to 57.8%), ciprofloxacin (from 49.5% to 39.6%), norfloxacin (from 46.8% to 37.3%), and amoxicillin/clavulanic acid (from 40.8% to 23.5%) decreased compared to before the ban. However, resistance to cephalothin (from 51.4% to 66.5%), cefepime (from 0.0% to 2.4%), and colistin (from 7.3% to 11.0%) had increased. We confirmed a high percentage of multidrug resistance before (95.0%) and after (96.6%) the ban on AGPs. The *AmpC* gene was the most prevalent from 2007 to 2017 (60.0%), followed by the *blaTEM* gene (55.5%). The *blaTEM* was prevalent before (2007–2011, 69.3%) and after (2012–2017, 49.2%) the ban on AGPs. These results provide data that can be used for the prevention and treatment of enteric colibacillosis.

## 1. Introduction

Weaned piglets are vulnerable to disease for many reasons such as changes in environmental conditions, a decline in maternal antibody titers, and various stresses. Presently, enteric colibacillosis, such as postweaning diarrhea and/or edema disease, is frequent in swine farms [1]. *Escherichia coli* (*E. coli*) in weaned piglets can result in serious economic losses due to diarrhea, growth retardation, and increased mortality [2].

Antimicrobials are used to treat colibacillosis in diarrheic weaned piglets. They play a significant role in the prevention and treatment of diseases and have also been used as a feed additive to promote swine growth [3]. However, the widespread and indiscriminate use of antimicrobials has resulted in the emergence of antimicrobial-resistant bacteria and the appearance of antibiotic residues in meat products [4]. Antimicrobial-resistant bacteria represent a threat to the successful treatment of disease in swine farms. As such, antibiotic growth promoters (AGPs) in feeds were banned in Korea in July 2011 [5].

Due to the emergence of antimicrobial resistance as a global problem, Denmark [6], Japan [7], and Canada [8] have started to formally monitor the state of antibiotic resistance. The antimicrobial-resistant profiles of pathogenic *E. coli* are changing based on geographical and temporal variations and previous exposures to antimicrobial agents [9]. Thus, devising control measures for colibacillosis in piggeries requires data regarding the prevalence of antimicrobial susceptibility [9].

Information on the antimicrobial resistance of pathogenic *E. coli* will be useful in establishing treatment and prevention strategies for colibacillosis in the swine industry. Although there have been many studies on the antimicrobial resistance of *E. coli* [10,11,12], few studies have examined annual trends in antimicrobial resistance profiles of *E. coli* over a decade. Additionally, there was little data comparing the changes in antimicrobial resistance profiles before and after the ban on AGPs. This study aimed to survey the antimicrobial resistance profiles of 690 pathogenic *E. coli* isolates obtained from Korean pigs with symptoms of enteric colibacillosis between 2007 and 2017, while assessing the change in antimicrobial resistance profiles before and after the ban on AGPs.

## 2. Results

### 2.1. Antimicrobial Susceptibility Test

The results of the antimicrobial susceptibility tests are shown in Table 1. We confirmed high resistance rates to tetracycline (598 isolates, 86.7%), ampicillin (586 isolates, 84.9%), streptomycin (589 isolates, 85.4%). On the other hand, the strains showed low resistance to cefepime (10 isolates, 1.4%) and colistin (68 isolates, 9.9%).

After the ban on AGPs, resistance to gentamicin (68.8% to 39.0%), neomycin (84.9% to 57.8%), ciprofloxacin (49.5% to 39.6%), norfloxacin (46.8% to 37.3%), and amoxicillin/clavulanic acid (40.8% to 23.5%) decreased with respect to that before the ban. There was a rapid decline in the resistance rates of those antimicrobials between 2012 to 2013 and 2014 to 2015. The resistance rates of gentamicin, ciprofloxacin, and norfloxacin in 2012–2013 were 62.4%, 59.6%, and 58.2%, respectively. However, those were decreased to 35.3%, 29.4%, and 27.1% in 2016–2017, respectively. On the other hand, the resistance rate of cephalothin (51.4% to 69.5%) increased, with an emergence of resistance to cefepime (0.0% to 2.1%) after the ban on AGPs. When we compared the resistance to colistin between 2012 to 2013 and 2014 to 2015 using a Chi-square test, there was a significant increase in resistance to colistin (7.1% to 19.9%, *p* < 0.01).

### 2.2. Multidrug Resistance

The results of the analysis of multidrug resistance rates are shown in Table 2. Before and after the ban on AGPs, the percentage of isolates resistant to seven subclasses were 22.9% and 21.0%, respectively, which were the most prevalent. Before the ban on AGPs, 1.4%, 6.4%, and 10.6% of isolates showed patterns of resistance to three, four, and eight antimicrobial subclasses, respectively. These rates increased to 4.0%, 9.3%, and 16.1% after the ban. However, rates of resistance to 6 and 10 subclasses decreased from 18.3% to 14.8% and from 11.9% to 7.4%, respectively. In terms of multidrug resistance for those with resistance to 3 or more subclasses of drugs among the 12 subclasses of drugs tested, 207 (95.0%) strains before the ban on AGPs, and 456 (96.6%) strains after the ban on AGPs, respectively, showed multidrug resistance.

### 2.3. Antimicrobial Resistance Genes

Table 3 shows the prevalence of antimicrobial resistance genes of *E. coli* isolated from diarrheic weaned piglets before and after the ban on AGPs. The *AmpC* gene was the most prevalent from 2007 to 2017 (414 isolates, 60.0%), followed by the *blaTEM* gene (383 isolates, 55.5%). Although the percentage decreased after the ban on AGPs, the *blaTEM* gene was still prevalent in 2007–2011 (69.3%) and 2012–2017 (49.2%). Before the ban on AGPs, only five isolates (2.3%) encoded the *blaSHV* gene. However, after the ban on AGPs, 29 isolates (6.1%) tested *blaSHV* positive. Additionally, there was the emergence of *blaCTX-M* group 2 after the ban on AGPs (0.0% to 1.7%). There was no isolate encoding the *mcr*-2 gene in this study. Rates of the *tetA* gene increased annually. From 2007 to 2011, there were 46 (21.1%) *tetA*-gene-encoding isolates. However, it increased to 32.4% (153 isolates) in 2012–2017.

## 3. Discussion

Antibiotics are used in intensive pig production systems to control infectious diseases. This widespread use is suspected to be a major cause of antimicrobial resistance [13]. To treat colibacillosis, antimicrobial agents including broad-spectrum-activity drugs, such as β-lactams and fluoroquinolones, are frequently used in veterinary medicine [14]. Usage of antimicrobial agents could be a cause for increasing antimicrobial resistance [15]. Thus, diseased animals might constitute an important reservoir of antimicrobial resistance [14].

In the previous study, we isolated 690 pathogenic *E. coli* strains from weaned piglets showing signs of enteric colibacillosis from 2007 to 2017, and investigated these isolates for adherence (F4, F5, F6, F18, F41, eae, paa, AIDA-I) and toxin (LT, STa, STb, Stx2e, EAST-I) genes [4]. Further, in this study, we tested antimicrobial resistance phenotypes and genotypes. We sought to provide data on the annual antimicrobial resistance profiles in Korean pig farms.

The frequencies of resistance to antimicrobials (gentamicin: 48.4%, neomycin: 66.4%, nalidixic acid 70.3%, ampicillin 84.9%, trimethoprim/sulfamethoxazole 60.9%, and tetracycline 86.7%) observed in this study are clearly higher than the EU average resistance figures (nalidixic acid 59.8%, ampicillin 58.0%, and tetracycline 47.1%), and the USA average resistance figures (gentamicin: 23.9%, neomycin 33.8%, ampicillin 68.1%, and trimethoprim/sulfamethoxazole 22.0%) [16,17].

Caution must be exercised when comparing such data because of the differences in methodologies used, particularly with the use of Clinical and Laboratory Standards Institute (CLSI) clinical breakpoints in this study compared with the use of epidemiological cut-off values (ECOFF’s) in the EFSA (European Food Safety Authority) report. The results of this study are based on multiple isolates from diseased piglets. In contrast, the EFSA data were gathered mainly from the national monitoring program based on the sampling of healthy porcine carcasses at slaughter with all samples derived from distinct epidemiological units. The differences in resistance data across countries are not completely unexpected, as intensive pig production throughout Europe operates to different standards and utilizes distinct management practices.

Additionally, the pathogenic *E. coli* strains isolated in this study exhibited high resistance to ampicillin (84.9%), tetracycline (86.7%), and streptomycin (85.4%). This result is similar to the results of studies published in Denmark [6], Japan [7], and Canada [8]. The comparisons of resistance rates to the antimicrobials tested revealed that the isolates were more frequently resistant to ampicillin, tetracycline, and streptomycin—drugs that have been extensively used in large quantities in Korea [18]. Similar results have been described in other Korean reports. Cho et al. reported that the rate of resistance to tetracycline was the highest (97.8%), followed by ampicillin (89.1%) [19]. Lim et al. also reported a high resistance rate of *E. coli* to tetracycline (76.1%), ampicillin (64.6%), and streptomycin (58.4%) [10]. However, the resistance rates reported in this study were higher than those reported by Lim et al. [10]. This might be due to the differences in the origins of the isolates. We isolated from weaned piglets showing symptoms of colibacillosis, however, Lim et al. isolated from healthy pigs. According to the Korean national antimicrobial resistance monitoring systems, pathogenic bacteria tend to be more resistant to antimicrobials than bacteria isolated from normal livestock [20]. Lim et al. [10] assessed the resistance rates of *E. coli* from normal livestock; whereas in this study, we tested the antimicrobial resistance of pathogenic *E. coli* encoding at least one or more virulence factors isolated from pigs with diarrhea.

After the ban on AGPs, resistance rates to gentamicin had dramatically decreased (68.8% to 39.0%). Additionally, there was the emergence of resistance to cefepime (0.0% to 2.1%) and an increase in resistance rates to cephalothin (51.4% to 69.5%) and colistin (7.3% to 11.0%). Antimicrobial resistance is dependent on the level of antimicrobial usage [4]. The sales for antimicrobial agents that are usually used for growth promotion in Korea decreased from 2010 to 2017. Antimicrobial classes, tetracyclines and aminoglycosides, sold as much as 283,865 kg and 58,975 kg in 2010, respectively. However, this decreased to 254,541 kg and 50,503 kg in 2017, respectively. On the other hand, the sales for cephalosporins and phenicols, which are frequently used to treat enteric diseases in swine, increased from 2010 (4980 kg, and 63,882 kg, respectively) to 2017 (11,312 kg, and 114,716 kg, respectively) [21]. These changes in sales for antimicrobial agents could affect the resistance of isolates. Due to the rare occurrence of bacteria with resistance to it, as well as a paucity of horizontal transmission of resistance mechanisms, colistin has been regularly used for the treatment of enteric colibacillosis [22]. Additionally, the World Health Organization (WHO) has classified colistin as one of the “Highest Priority Critically Important Antimicrobials” in humans [23]. Recently, the plasmid-mediated colistin resistance gene, *mcr* was reported in Korea [24,25]. The observed increase in resistance rates could be attributed to this *mcr* gene. Increased resistance to colistin could pose serious problems not only in veterinary medicine but also in public health. Restrictions on the use of colistin are required to reduce resistance rates.

We confirmed a high frequency of multidrug resistance before (207 isolates, 95.0%) and after (456 isolates, 96.6%) the ban on AGPs. Due to the different types of antimicrobial tested, it is hard to directly compare multidrug resistance rates in comparison to the multidrug resistance rates (30.9%) of pigs with *E. coli* of US origin [26], the multidrug resistance rates of Korean piglets were very high. Since the regulation of the use of antimicrobials is not as strict in Korea as it is in other developed countries, it is considered that the use of antimicrobials by nonspecialists such as livestock workers and not veterinarians might be the cause of this phenomenon [18].

*E. coli* develop resistance mechanisms by using instructions provided by their DNA. Often, antimicrobial resistance genes are found within plasmids. This means that some bacteria can share their antimicrobial resistance genes and make other bacteria become resistant [3]. Extended spectrum β-lactamases (ESBLs) have been reported worldwide, most frequently in Enterobacteriaceae. ESBLs and *AmpC* are plasmid-encode, which are capable of inactivating a large number of beta-lactam antibiotics [15]. Additionally, colistin has been regularly used for the treatment of enteric colibacillosis such as postweaning diarrhea and edema disease due to the rare existence of resistant bacteria and lacking horizontal transmission mechanisms. However, very recently, the plasmid-mediated colistin resistance gene, *mcr* was found in Korea [5]. 

The *AmpC* gene encodes cephalosporinases and gives rise to serious therapeutic challenges in veterinary medicine. In this study, 414 isolates (60.0%) were positive for *AmpC* gene. β-lactam resistance in *E. coli* generally occurs as a result of the deregulation of the putative *AmpC* gene or the acquisition of a mobile genetic element containing an *AmpC* gene [15]. Consequently, high rates of ampicillin-resistant isolates are to be expected. In this study, we found that there was a high rate of resistance to ampicillin (84.9%).

Both TEM and SHV enzymes belong to the class A family of β-lactamases and are widely disseminated among the *Enterobacteriaceae* from veterinary sources [27] In this study, *blaTEM* was identified in over half the porcine *E. coli* isolates. However, *blaSHV* was detected in only 4.9% of isolates. There was also a decrease in the frequency of *blaTEM* after the ban on AGPs. TEM enzymes often co-exist with CTX-M enzymes in bacteria of animal origin [28]. However, the number of *blaCTX-M* group-positive isolates in this study was low (group 1: 13 isolates, group 2: 8 isolates, group 9: 18 isolates, total: 39 isolates). These results are of interest considering the current epidemiology of these genes.

We also confirmed that the antimicrobial resistance varied according to regions (Appendix A). Resistance to chloramphenicol, gentamicin, neomycin, nalidixic acid, ciprofloxacin, and trimethoprim/sulfamethoxazole was higher in the northern farms than middle and southern farms after the ban on AGPs. The antimicrobial resistance phenotypes, and the multidrug resistance rates and prevalence of the *tetA* gene increased after the ban on AGPs in the northern farms, unlike in the middle and southern farms. This result is probably due to the fact that more antimicrobials were used in the northern farms than in the middle and southern farms. From 2001 in Korea, sales of antimicrobials for domestic livestock products and fisheries by use, breed, and antimicrobials were analyzed by a comprehensive management system created by the Korean Animal Health Products Association [20]. However, the analyses were not carried out by region [10]. The WHO/FAO/OIE stresses that overall sales, livestock, annual, periodic, and regional sales data for the use of antimicrobials that can be compared internationally and shared together should be investigated in a standardized way and the results should be expressed in standardized units [29]. Proper risk assessment, guidelines, and policy decisions for antimicrobial resistance management require data on the use of antimicrobials in each region. Therefore, in order to obtain a more accurate use of antimicrobials, it will be necessary to classify antimicrobials used in livestock production in accordance with the International Standards Classification Act and to develop a surveillance system that can further refine the methods of investigation (such as route of administration, breeding, breeding stage, and disease), to enter all prescribed antimicrobial agents, and to train clinical veterinarians.

In this study, we analyzed and compared the antimicrobial resistance phenotypes and genotypes of *E. coli* isolated from Korean diarrheic weaned piglets before and after the ban on AGPs. The trend in our findings suggests a decrease in resistance to gentamicin, neomycin, ciprofloxacin, norfloxacin, and amoxicillin/clavulanic acid after the ban on AGPs. However, resistance to cephalothin, cefepime, and colistin increased. Additionally, there was still a high frequency of multidrug-resistant isolates. Among the tested antimicrobial resistance genes, *AmpC* and *blaTEM* genes were the most prevalent. After the ban on AGPs, the frequency of the *AmpC* gene increased. On the other hand, the frequency of the *blaTEM* gene decreased. These results provide data that can be used for the prevention and treatment of enteric colibacillosis, as well as important data for assessing the impact of banning AGPs on the antimicrobial resistance profiles of *E. coli* isolates. Further studies are needed to determine the specific association of exposures to antimicrobial agents with antimicrobial resistance profiles.

## 4. Materials and Methods

### 4.1. Escherichia coli Isolates

Between 2007 and 2017, 690 *E. coli* isolates were obtained from weaned piglets showing symptoms of enteric colibacillosis and/or edema disease. The sampled farms consisted of 150 different pig herds (50 to 100 sows per herd) located in 3 areas: the northern (35 farms encompassing the Gangwon, Gyeonggi, and Incheon provinces), middle (46 farms, Chungbuk and Chungnam provinces), and southern (69 farms, Chonbuk, Chonnam, Gyeongbuk, and Gyeongnam provinces) areas of South Korea. The strains were not collected repeatedly from the same farm. The aseptically collected intestinal contents and feces were inoculated on a MacConkey agar (Becton Dickinson, Sparks, MD, USA) and blood agar (Asan Pharmaceutical, Seoul, Korea). After overnight incubation at 37 °C, only pure or nearly pure cultured colonies were selected and transferred to blood agar. Suspected colonies were identified as *E. coli* using the VITEK 2 GN ID card via VITEK II system (bioMéreiux, Marcy l’Etoile, France). The isolates were stored in 20% glycerol at −70 °C for further experimentation.

### 4.2. Antimicrobial Susceptibility Test

The following 16 antimicrobials were selected following the marketing amounts for animal use in Korea: gentamicin (10 μg), streptomycin (10 μg), neomycin (30 μg), ampicillin (10 μg), amoxicillin/clavulanic acid (20/10 μg), cephalothin (30 μg), cefoxitin (30 μg), cefazolin (30 μg), cefepime (30 μg), nalidixic acid (30 μg), ciprofloxacin (5 μg), norfloxacin (10 μg), sulfamethoxazole/trimethoprim (23.75/1.25 μg), chloramphenicol (30 μg), colistin (10 μg), and tetracycline (30 μg). Each antimicrobial disc was purchased from Becton-Dickinson (Sparks, MD, USA). Antimicrobial susceptibility testing was carried out using the Kirby Bauer disk diffusion method [30]. The isolates were inoculated on Mueller-Hinton agar (Becton-Dickinson). The antimicrobial discs were dropped on the agar and incubated at 37 °C for 18 h. *Escherichia coli* ATCC 25922 was used for quality control of the experiment. The interpretation of zone of inhibition was determined according to the CLSI standards *Enterobacteriaceae* breakpoints [31]. Intermediate isolates were grouped with resistant isolates. CLSI classified antimicrobial agents by class including several subclasses [31]. We categorized antimicrobial agents according to CLSI antimicrobial subclasses. Strains resistant to 3 or more CLSI subclasses of drugs were considered as multidrug-resistant strains.

### 4.3. Antimicrobial Resistance Genes

The *E. coli* genes for antimicrobial resistance were amplified by polymerase chain reaction (PCR) analysis. Bacterial colonies were suspended in 200 μL of distilled water and boiled for 10 min. After centrifugation at 8000× *g*, the supernatant was used as a template for PCR. We tested colistin-resistant *mcr* genes [5], tetracycline-resistant *tetA* genes [32], ampicillin-resistant *AmpC* genes [33], and extended-spectrum β-lactamase: *blaTEM*, *blaSHV*, *blaOXA*, *blaCTX-M* group 1, group 2, and group 9 [34], according to previously-described protocols (Appendix A). Bacterial colonies were suspended in 200 μL of distilled water and boiled for 10 min. After centrifugation at 8000× *g*, the supernatant was used as a template. The reaction volume, 20 μL, was composed of 2× EmeraldAmp Master Mix (Takara, Otsu, Japan), 2 μM of each primer, and 2 μL of template DNA. PCR product was electrophoresed on 2% agarose gel using Mupid-exU AD140 (Takara), stained with Ethidium bromide, and visualized on a UV trans-illuminator.

### 4.4. Statistical Analysis

All statistical analyses were performed using SPSS version 12.0 program (SPSS inc., Chicago, IL, USA). To compare antimicrobial resistance before and after the ban on AGPs, chi-square test was performed.

## Figures and Tables

**Table 1 antibiotics-09-00755-t001:** Antimicrobial-resistant pathogenic *Escherichia coli* isolates (%) from weaned piglets with diarrhea in Korea from 2007 to 2017.

Antimicrobial		Before the Ban on AGPs			After the Ban on AGPs				Total (2007–2017) (*n* = 690)	Differences in before and after Ban on AGPs ^2^
Subclasses	Agents ^1^	2007–2009 (*n* = 118)	2010–2011 (*n* = 100)	Subtotal (2007–2011) (*n* = 218)	2012–2013 (*n* = 141)	2014–2015 (*n* = 161)	2016–2017 (*n* = 170)	Subtotal (2012–2017) (*n* = 472)		
**Aminoglycosides**	**GM ****	84 (71.2)	66 (66.0)	150 (68.8)	88 (62.4)	36 (22.4)	60 (35.3)	184 (39.0)	334 (48.4)	(29.8)
	**S**	99 (83.9)	85 (85.0)	184 (84.4)	122 (86.5)	131 (81.4)	152 (89.4)	405 (85.8)	589 (85.4)	(−1.4)
	**N ****	100 (84.7)	85 (85.0)	185 (84.9)	95 (67.4)	89 (55.3)	89 (52.4)	273 (57.8)	458 (66.4)	(27.1)
**Cephalosporin I**	**CF ****	66 (55.9)	46 (46.0)	112 (51.4)	95 (67.4)	120 (74.5)	113 (66.5)	328 (69.5)	440 (63.8)	(−18.1)
	**CZ**	23 (19.5)	18 (18.0)	41 (18.8)	30 (21.3)	55 (34.2)	22 (12.9)	107 (22.7)	148 (21.4)	(−3.9)
**Cephalosporin IV**	**FEP^*^**	0 (0.0)	0 (0.0)	0 (0.0)	1 (0.7)	5 (3.1)	4 (2.4)	10 (2.1)	10 (1.4)	(−2.1)
**Cephamycin**	**FOX**	20 (16.9)	13 (13.0)	33 (15.1)	24 (17.0)	31 (19.3)	6 (3.5)	61 (12.9)	94 (13.6)	(2.2)
**Quinolones**	**NA**	97 (82.2)	66 (66.0)	163 (74.8)	108 (76.6)	109 (67.7)	105 (61.8)	322 (68.2)	485 (70.3)	(6.6)
**Fluoroquinolone**	**CIP ***	67 (56.8)	41 (41.0)	108 (49.5)	84 (59.6)	53 (32.9)	50 (29.4)	187 (39.6)	295 (42.8)	(9.9)
	**NOR ***	63 (53.4)	39 (39.0)	102 (46.8)	82 (58.2)	48 (29.8)	46 (27.1)	176 (37.3)	278 (40.3)	(9.5)
**Aminopenicillin**	**AM**	103 (87.3)	86 (86.0)	189 (86.7)	116 (82.3)	131 (81.4)	150 (88.2)	397 (84.1)	586 (84.9)	(2.6)
**BL/BLI ^3^**	**AMC ****	36 (30.5)	53 (53.0)	89 (40.8)	54 (38.3)	43 (26.7)	14 (8.2)	111 (23.5)	200 (29.0)	(17.3)
**FPI ^4^**	**SXT**	92 (78.0)	41 (41.0)	133 (61.0)	94 (66.7)	97 (60.2)	96 (56.5)	287 (60.8)	420 (60.9)	(0.2)
**Phenicols**	**C**	104 (88.1)	90 (90.0)	194 (89.0)	125 (88.7)	130 (80.7)	154 (90.6)	409 (86.7)	603 (87.4)	(2.3)
**Polymyxins**	**CL**	9 (7.6)	7 (7.0)	16 (7.3)	10 (7.1)	32 (19.9)	10 (5.9)	52 (11.0)	68 (9.9)	(−3.7)
**Tetracyclines**	**TE ****	111 (94.1)	90 (90.0)	201 (92.2)	119 (84.4)	127 (78.9)	151 (88.8)	397 (84.1)	598 (86.7)	(8.1)

^1^ Significant differences between before (2007–2011) and after (2012–2017) the ban on antibiotic growth promoters in feed were expressed as * (*p* < 0.05) and ** (*p* < 0.01); GM: gentamicin, S: streptomycin, N: neomycin, CF: cephalothin, CZ: cefazolin, FEP: cefepime, FOX: cefoxitin, NA: nalidixic acid, CIP: ciprofloxacin, NOR: norfloxacin, AM: ampicillin, AMC: amoxicillin/clavulanic acid, SXT: trimethoprim/sulfamethoxazole, C: chloramphenicol, CL: colistin, TE: tetracycline. ^2^ Difference of the subtotals before (2007–2011) minus after (2012–2017) the ban on antibiotic growth promoters in feed. ^3^ BL/BLI: β-lactam/β-lactamase inhibitor combination. ^4^ FPI: Folate pathway inhibitors.

**Table 2 antibiotics-09-00755-t002:** Multidrug-resistant pathogenic *Escherichia coli* isolates (%) from weaned piglets with diarrhea in Korea from 2007 to 2017.

Antimicrobial Subclass ^1^	Before the Ban on AGPs			After the Ban on AGPs				Total (2007–2017) (*n* = 690)	Differences in before and after Ban on AGPs ^2^
	2007–2009 (*n* = 118)	2010–2011 (*n* = 100)	Subtotal (2007–2011) (*n* = 218)	2012–2013 (*n* = 141)	2014–2015 (*n* = 161)	2016–2017 (*n* = 170)	Subtotal (2012–2017) (*n* = 472)		
**0 subclass ****	4 (3.4)	0 (0.0)	4 (1.8)	0 (0.0)	0 (0.0)	0 (0.0)	0 (0.0)	4 (0.6)	(1.8)
**1 subclass**	2 (1.7)	0 (0.0)	2 (0.9)	3 (2.1)	1 (0.6)	2 (1.2)	6 (1.3)	8 (1.2)	(−0.4)
**2 subclasses**	0 (0.0)	5 (5.0)	5 (2.3)	4 (2.8)	4 (2.5)	2 (1.2)	10 (2.1)	15 (2.2)	(0.2)
**3 subclasses**	1 (0.8)	2 (2.0)	3 (1.4)	3 (2.1)	10 (6.2)	6 (3.5)	19 (4.0)	22 (3.2)	(−2.6)
**4 subclasses**	2 (1.7)	12 (12.0)	14 (6.4)	7 (5.0)	14 (8.7)	23 (13.5)	44 (9.3)	58 (8.4)	(−2.9)
**5 subclasses**	10 (8.5)	21 (21.0)	31 (14.2)	21 (14.9)	17 (10.6)	34 (20.0)	72 (15.3)	103 (14.9)	(−1.1)
**6 subclasses**	27 (22.9)	13 (13.0)	40 (18.3)	17 (12.1)	19 (11.8)	34 (20.0)	70 (14.8)	110 (15.9)	(3.5)
**7 subclasses**	28 (23.7)	22 (22.0)	50 (22.9)	32 (22.7)	26 (16.1)	41 (24.1)	99 (21.0)	149 (21.6)	(1.9)
**8 subclasses**	17 (14.4)	6 (6.0)	23 (10.6)	23 (16.3)	36 (22.4)	17 (10.0)	76 (16.1)	99 (14.3)	(−5.5)
**9 subclasses**	11 (9.3)	8 (8.0)	19 (8.7)	11 (7.8)	19 (11.8)	7 (4.1)	37 (7.8)	56 (8.1)	(7.9)
**10 subclasses**	16 (13.6)	10 (10.0)	26 (11.9)	18 (12.8)	13 (8.1)	4 (2.4)	35 (7.4)	61 (8.8)	(4.5)
**11 subclasses**	0 (0.0)	1 (1.0)	1 (0.5)	2 (1.4)	2 (1.2)	0 (0.0)	4 (0.8)	5 (0.7)	(−0.3)
**Multi-resistant (≥3 Subclasses)**	112 (94.9)	95 (95.0)	207 (95.0)	134 (95.0)	156 (96.9)	166 (97.6)	456 (96.6)	663 (96.1)	(−2.6)

^1^ Significant differences between before (2007–2011) and after (2012–2017) the ban on antibiotic growth promoters in feed were expressed as ** (*p* < 0.01) Antimicrobial subclass are defined by the Clinical and Laboratory Standards Institute. ^2^ Difference of the subtotals before (2007–2011) minus after (2012–2017) the ban on antibiotic growth promoters in feed.

**Table 3 antibiotics-09-00755-t003:** Antimicrobial resistance genes (%) of pathogenic *Escherichia coli* from weaned piglets with diarrhea in Korea from 2007 to 2017.

Antimicrobial Resistance GENES ^1^	Before the Ban on AGPs			After the Ban on AGPs				Total (2007–2017) (*n* = 690)	Differences in before and after Ban on AGPs ^2^
	2007–2009 (*n* = 118)	2010–2011 (*n* = 100)	Subtotal (2007–2011) (*n* = 218)	2012–2013 (*n* = 141)	2014–2015 (*n* = 161)	2016–2017 (*n* = 170)	Subtotal (2012–2017) (*n* = 472)		
*blaTEM* **	79 (66.9)	72 (72.0)	151 (69.3)	62 (44.0)	65 (40.4)	105 (61.8)	232 (49.2)	383 (55.5)	(20.1)
*blaSHV* *	2 (1.7)	3 (3.0)	5 (2.3)	11 (7.8)	16 (9.9)	2 (1.2)	29 (6.1)	34 (4.9)	(−3.8)
*blaOXA*	10 (8.5)	14 (14.0)	24 (11.0)	19 (13.5)	18 (11.2)	9 (5.3)	46 (9.7)	70 (10.1)	(1.3)
*blaCTX-M* group 1	0 (0.0)	4 (4.0)	4 (1.8)	2 (1.4)	5 (3.1)	2 (1.2)	9 (1.9)	13 (1.9)	(−0.1)
*blaCTX-M* group 2	0 (0.0)	0 (0.0)	0 (0.0)	0 (0.0)	8 (5.0)	0 (0.0)	8 (1.7)	8 (1.2)	(−1.7)
*blaCTX-M* group 9	0 (0.0)	4 (4.0)	4 (1.8)	3 (2.1)	9 (5.6)	2 (1.2)	14 (3.0)	18 (2.6)	(−1.2)
*mcr*-1	0 (0.0)	1 (1.0)	1 (0.5)	3 (2.1)	3 (1.9)	1 (0.6)	7 (1.5)	8 (1.2)	(−1.0)
*mcr*-2	0 (0.0)	0 (0.0)	0 (0.0)	0 (0.0)	0 (0.0)	0 (0.0)	0 (0.0)	0 (0.0)	(0.0)
*mcr*-3	1 (0.8)	2 (2.0)	3 (1.4)	5 (3.5)	0 (0.0)	0 (0.0)	5 (1.1)	8 (1.2)	(0.3)
*AmpC* **	54 (45.8)	43 (43.0)	97 (44.5)	88 (62.4)	105 (65.2)	124 (72.9)	317 (67.2)	414 (60.0)	(−22.7)
*tetA* **	20 (16.9)	26 (26.0)	46 (21.1)	38 (27.0)	60 (37.3)	55 (32.4)	153 (32.4)	199 (28.8)	(−11.3)

^1^ Significant differences between before (2007–2011) and after (2012–2017) the ban on antibiotic growth promoters in feed were expressed as * (*p* < 0.05) and ** (*p* < 0.01). ^2^ Difference of the subtotals before (2007–2011) minus after (2012–2017) the ban on antibiotic growth promoters in feed.

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
