# Peer review of "Antimicrobial Resistance Profiles of Escherichia coli from Diarrheic Weaned Piglets after the Ban on Antibiotic Growth Promoters in Feed"

_antibiotics, 2020, doi:10.3390/antibiotics9110755_

Round 1

Reviewer 1 Report

This study gives an overview on the effect of the cessation of the use of antibiotics as growth promoters in pigs on a collection of E. coli isolates collected both before and after this cessation came into effect. The study looks at phenotypic antibiotics resistance (disk diffusion testing) and looks at the presence of selected antimicrobial resistance genes

The paper is well written and with corrections will contributes greatly to the information concerning the effect of the cessation of antibiotic usage as growth promoters.

Major Points

Through the text the authors state that for some antibiotics there is a decrease in the level of antimicrobial resistance following the ending of the usage of antibiotics as growth promoters (an expected result) however for other antibiotics there is an increase in resistance levels. This can also be seen in an increase in some levels of multi-drug resistance. The authors have not given an explanation/ hypothesis for why this is that case. This must be added to paper.

Do the authors have any information (on a country/regional/farm basis?) on what was the level of usage of antibiotics before the ban on their usage as growth promoters and the level of usage afterwards? If this information was added it could add greatly to the paper.

Minor Points

  1. In Table 1 Gentamicin is referred to as GM however in the footnotes it is G. Please correct.
  2. In Table 1 what does the superscript on AMC and SXT refer to? 
  3. Can the authors add a description of "sub- classes" of antibiotic to the paper for clarity?
  4. Lines 128 -133 are repetition of lines 110 - 115. Please remove.
  5. Line 137 states "This might be due to the differences in the origins of the isolates". What was this difference?
  6. Line 159 - 166. The point that the authors are trying to get here is muddled. Line 159-160 should have their own paragraph and points raised should be discussed further. Then the paragraph discussing AmpC can follow.
  7. In "4.1 Escherichia coli isolates" What VITEK card was used in the species identification?
  8. In "4.2. Antimicrobial susceptibility test" what controls were used when testing the E. coli isolate antibiotic resistance pattern?
  9. Why were the antibiotics used in this study chosen?
  10. Presumably results were compared to Enterobacterales breakpoints? Please clarify.
  11. Please add a reference to CLSI methods and breakpoints used.
  12. In"4.3. Antimicrobial resistance genes" the authors must add the PCR Primers used and the PCR recipe and program in this study to amplify Antibiotic resistance genes (This could be added to supplementary data)
  13. How were these genes chosen for analysis?

Author Response

This study gives an overview on the effect of the cessation of the use of antibiotics as growth promoters in pigs on a collection of E. coli isolates collected both before and after this cessation came into effect. The study looks at phenotypic antibiotics resistance (disk diffusion testing) and looks at the presence of selected antimicrobial resistance genes

The paper is well written and with corrections will contributes greatly to the information concerning the effect of the cessation of antibiotic usage as growth promoters.

Major Points

Through the text the authors state that for some antibiotics there is a decrease in the level of antimicrobial resistance following the ending of the usage of antibiotics as growth promoters (an expected result) however for other antibiotics there is an increase in resistance levels. This can also be seen in an increase in some levels of multi-drug resistance. The authors have not given an explanation/ hypothesis for why this is that case. This must be added to paper.

Do the authors have any information (on a country/regional/farm basis?) on what was the level of usage of antibiotics before the ban on their usage as growth promoters and the level of usage afterwards? If this information was added it could add greatly to the paper.

Answer: According to reviewer’s opinion, we added information on sales of antibiotics before and after
the ban on antibiotic growth promoters in feed on line 151-158.

Line 151-158: Antimicrobial resistance is depending on the level of antimicrobial usage [4]. The sales for antimicrobial agents that usually used for growth promotion in Korea decreased from 2010 to 2017. Antimicrobial classes, tetracyclines and aminoglycosides, were sold 283,865 kg and 58,975 kg in 2010, respectively. However, those decreased to 254,541 kg and 50,503 kg in 2017, respectively. On the other hand, the sales for cephalosporins and phenicols, which are frequently used to treat enteric diseases in swine, increased from 2010 (4,980 kg, and 63,882 kg, respectively) to 2017 (11,312 kg, and 114,716 kg, respectively) [21]. These changes in sales for antimicrobial agents could affect the resistance of isolates.

Minor Points

  1. In Table 1 Gentamicin is referred to as GM however in the footnotes it is G. Please correct.
    Answer: We corrected errata G to GM in Table 1.
    Line 94: G → GM
    Supplementary data line 3: G → GM

  2. In Table 1 what does the superscript on AMC and SXT refer to? 
    Answer: We deleted the superscript on AMC and SXT in Table 1.
    Line 100: Table 1 – AMC1, SXT2 → AMC, SXT
    Supplementary data line 1: Table 1 - AMC1, SXT2 → AMC, SXT

  3. Can the authors add a description of "sub- classes" of antibiotic to the paper for clarity?
    Answer: We added description for subclass of antibiotics on Table 1 and line 253-254.
    Line 253-254: CLSI classified antimicrobial agents by class including several subclasses [31].
      We categorized antimicrobial agents according to CLSI antimicrobial subclasses.

  4. Lines 128 -133 are repetition of lines 110 - 115. Please remove.
    Answer: We deleted lines 110-115.

  1. Line 137 states "This might be due to the differences in the origins of the isolates". What was this difference?
    Answer: We added line 143-144 to explain the differences in the origins of the isolates.
    Line 143-144: This might be due to the differences in the origins of the isolates. We isolated
      from weaned piglets showing symptoms of colibacillosis however, Lim SK et al. isolated from
      healthy pigs.

  2. Line 159 - 166. The point that the authors are trying to get here is muddled. Line 159-160 should have their own paragraph and points raised should be discussed further. Then the paragraph discussing AmpC can follow.
    Answer: For reader friendly, we revised these, and moved the line 159-166 to line 109-113.
    Line 109-113: To treat colibacillosis, antimicrobial agents including broad-spectrum-activity
      drugs, such as β-lactams and fluoroquinolones, are frequently used in veterinary medicine
      [14]. Usage of antimicrobial agents could be a cause for increasing antimicrobial resistance
      [15]. Thus, diseased animals might constitute an important reservoir of antimicrobial
      resistance [14].

  3. In "4.1 Escherichia coli isolates" What VITEK card was used in the species identification?
    Answer: We added the information about VITEK card to line 238.
    Line 238: Suspected colonies were identified as E. coli using the VITEK 2 GN ID card via VITEK
      â…¡ system (bioMéreiux, France).

  4. In "4.2. Antimicrobial susceptibility test" what controls were used when testing the E. coli isolate antibiotic resistance pattern?
    Answer: We added information about quality control of the experiments on line 249-251. (ATCC
      25922)
    Line 249-251: Escherichia coli ATCC 25922 was used for quality control of the experiment.

  5. Why were the antibiotics used in this study chosen?
    Answer: These antibiotics are frequently used for animal use in Korea. We added this
      information on line 242-243.
    Line 242-243: The following 16 antimicrobials were selected following the marketing amounts
      for animal use in Korea:  

  6. Presumably results were compared to Enterobacterales breakpoints? Please clarify.
    Answer: We used Enterobacteriaceae breakpoints, and added this information on line 251-252.
    Line 251-252: The interpretation of zone of inhibition was determined according to the CLSI
      standards Enterobacteriaceae breakpoints [31].  

  1. Please add a reference to CLSI methods and breakpoints used.
    Answer: We added reference [31] for CLSI methods and breakpoints.
    Line 350-352: 31. Clinical and Laboratory Standards Institute. CLSI Supplement M100.
     Performance Standards for Antimicrobial Susceptibility Testing, 28th ed.; Clinical and
       Laboratory Standards Institute: Wayne, PA, USA, 2018.

  2. In"4.3. Antimicrobial resistance genes" the authors must add the PCR Primers used and the PCR recipe and program in this study to amplify Antibiotic resistance genes (This could be added to supplementary data)
    Answer: We added information for PCR on Supplementary data Table S4.
    Supplementary data line 17: Table S4

  3. How were these genes chosen for analysis?
    Answer: We added the importance of antimicrobial resistance genes on line 173-181.
    Line 173-181: E. coli develop resistance mechanisms by using instruction provided by their
      DNA. Often, antimicrobial resistance genes are found within plasmids. This means that some
      bacteria can share their antimicrobial resistance genes and make other bacteria become
      resistant [3]. Extended spectrum β lactamases (ESBLs) have been reported worldwide, most
      frequently in Enterobacteriaceae. ESBLs and AmpC are plasmid-encode, which are capable
      of inactivating a large number of beta-lactam antibiotics [15]. Also, colistin has been regularly
      used for the treatment of enteric colibacillosis such as postweaning diarrhea and edema
      disease due to the rare existence of resistant bacteria and lacking horizontal transmission
      mechanisms. However, very recently, the plasmid-mediated colistin resistance gene, mcr
      was found in Korea [5].

Reviewer 2 Report

Abstract

(…the ban on AGPs)

Please give the full meaning (ie antibiotic growth promoters), since this is the first time it appears in text

Line 44-45

(…based on geographical and ……)

Please give specific references to support these points

Lines 194-204 and Table3

The introduction of a Figure could give a quick and helpful understanding of research findings before and after the ban of AGPs

Section 4 Materials and methods

Please give a short description about disk diffusion (4.2) method of phenotype analysis

Section Statistical Analysis

Please provide a relevant section explaining the software packages, methodology, variables relations, etc, used for data analysis

Author Response

  1. Abstract (…the ban on AGPs)
    Please give the full meaning (ie antibiotic growth promoters), since this is the first time it appears in text
    Answer: We revised the line 14-15 according to reviewer’s opinion.
    Line 14-15: while assessing the change in antimicrobial resistance profiles before and after
    the ban on antibiotic growth promoters (AGPs).

  2. Line 44-45 (…based on geographical and ……)
    Please give specific references to support these points
    Answer: The reference for supporting these points was [9]. For reader friendly, we added same reference to line 45.
    Line 45: The antimicrobial resistance profiles of pathogenic coli are changing based on
    geographical and temporal variations and previous exposures to antimicrobial agents [9].  

  3. Lines 194-204 and Table3
    The introduction of a Figure could give a quick and helpful understanding of research findings before and after the ban of AGPs
    Answer: Thank you for your suggestion. However, we think the table is more appropriate to
    give a helpful understanding of research findings. The Table 3 expresses too many kinds
      of time-track, and even more, some of those are overlap (2007-2008, 2010-2011, 2007-
      2011, 2012-2013, 2014-2015, 2016-2017, 2012-2017, and 2007-2017).
            For a quick and helpful understanding, we added column to show the difference of
      subtotals before minus after ban on AGPs on Table 1, Table 2, and Table 3.    

  4. Section 4 Materials and methods
    Please give a short description about disk diffusion (4.2) method of phenotype analysis
    Answer: We added a short description for disk diffusion method on line 249-252.
    Line 249-252: The isolates were inoculated on Mueller-Hinton agar (BD, US, USA). The
    antimicrobial discs were dropped on the agar and incubated at 37oC for 18 hours.
      Escherichia coli ATCC 25922 was used for quality control of the experiment. The
      interpretation of zone of inhibition was determined according to the CLSI standards
      Enterobacteriaceae breakpoints [31].

  5. Section Statistical Analysis
    Please provide a relevant section explaining the software packages, methodology, variables relations, etc, used for data analysis
    Answer: We added methods for statistical analysis on line 270-273.
    Line 270-273: All statistical analyses were performed using SPSS version 12.0 program
    (SPSS, Chicago, IL, USA). To compare antimicrobial resistance before and after the ban
      on AGPs, chi-square test was performed.

Reviewer 3 Report

line 14: abbreviation AGP should be introduced.

line 17 ff: missing spaces before brackets

line 23: what does ratio mean? Do you mean percentage?

line 23 ff: sentence must be rephrased since “after the ban” is period 2012-2017 but the way it is written it looks like 2007-2011 is “after the ban” (e.g. “…blaTEM was still prevalent in 2012-2017(49.2%) as compared to 2007-2011(69.3%).”)

line 32: Escherichia coli (E. coli)

line 35: It would be more appropriate to talk about “antibiotics” instead of “antimicrobials”, especially in sections like 2.1 where antibiotics – not antimicrobials – were tested.

line 66: …those were decreased to … in 2014-2015?

line 68: can statistics be done on these numbers? Otherwise avoid calling the increase “significant”.

line 71: Table 1: inconsistency between abbreviation GM and G (table vs. legend). Also, what are the superscript numbers in AMC and SXT? Finally, instead (or additionally) to showing “total 2007-2017” it would be informative to have a column with the difference of the subtotals before minus after ban. That way it would be visible directly which resistances decreased most (and which increased) after vs. before the ban.

line 79: patterns of resistance? Rather “percentage of isolates resistant to 7 subclasses”. In addition, it is more common to define multidrug resistance based on resistance to different classes (or for a more clinically relevant context “categories” as defined by Magiorakos et aline 2012). I could not find a CLSI definition of MDR, if there is one which uses the mentioned subclasses, Table 2 can remain as it is. Otherwise, consider revising.

line 94: better “percentage”, not ratio

line 117: Again, avoid “significantly” when there is no statistical result. Better write “clearly” or similar.

line 119 ff: why compare to Europe (only), when you used CLSI? Is there no US data?

line 129 ff: this passage is like a repetition of what was already mentioned in line 110 ff. Please omit here or before.

line 141: were your isolates screened for pathogenicity factors after isolation or was it assumed they are pathogenic because they originated from diarrheic piglets? If it was confirmed please mention it explicitly. Otherwise, pathogenicity is only assumed.

line 155: check phrasing “with E. coli with of US origin”. In addition, did the US study use the same definition of MDR as was used in this study? I also miss a citation here for these 51.7%

line 168: put Enterobacteriaceae in italics

Author Response

  1. line 14: abbreviation AGP should be introduced.
    Answer: We revised the line 14-15 according to reviewer’s opinion.
    Line 14-15: while assessing the change in antimicrobial resistance profiles before and after
    the ban on antibiotic growth promoters (AGPs).

  2. line 17 ff: missing spaces before brackets
    Answer: We have incorporated your suggestion throughout the manuscript.

  3. line 23: what does ratio mean? Do you mean percentage?
    Answer: We revised ratio to percentage on line 21, 73, and 85, 89.
    Line 21: ratio → percentage
    Line 73: ratio → percentage
    Line 85: ratio → percentage

  4. line 23 ff: sentence must be rephrased since “after the ban” is period 2012-2017 but the way it is written it looks like 2007-2011 is “after the ban” (e.g. “…blaTEMwas still prevalent in 2012-2017(49.2%) as compared to 2007-2011(69.3%).”)
    Answer: We revised this sentence on line 23-24.
    Line 23-24: The blaTEM was prevalent before (2007-2011, 69.3%) and after (2012-2017,
      2%) the ban on AGPs.

  5. line 32: Escherichia coli( coli)
    Answer: We revised Escherichia (E.) coli to Escherichia coli (E. coli) on line 32.
    Line 32: Escherichia (E.) coliEscherichia coli (E. coli)

  6. line 35: It would be more appropriate to talk about “antibiotics” instead of “antimicrobials”, especially in sections like 2.1 where antibiotics – not antimicrobials – were tested.
    Answer: Thank you for this suggestion. However, by strict definition, the word “antibiotic”
    refers to substance produced by microorganisms that act against another microorganisms.
      Thus, antibiotics do not include antimicrobial substances that are synthetic (quinolones), or
      semisynthetic (amoxicillin). So, I think “antimicrobials” is more appropriate.

  7. line 66: …those were decreased to … in 2014-2015?
    Answer: We revised this sentence on line 64-66.
    Line 64-66: The resistance rates of gentamicin, ciprofloxacin, and norfloxacin in 2012-2013
    were 62.4%, 59.6%, and 58.2%, respectively. However, those were decreased to 35.3%,
      4%, and 27.1% in 2016-2017, respectively.

  8. line 68: can statistics be done on these numbers? Otherwise avoid calling the increase “significant”.
    Answer: We compared antimicrobial resistance before and after the ban on AGPs using chi-square test. We added methods for statistical analysis on line 261-264, and revised the line 68-70, Table 1, Table 2, and Table 3.
    Line 68-70: When we compared the resistance to colistin between 2012-2013 and 2014-2015
    using chi-square test, there was a significant increase in resistance to colistin (7.1% to
      9%, P < 0.01).
    Line 261-264: All statistical analyses were performed using SPSS version 12.0 program
      (SPSS, Chicago, IL, USA). To compare antimicrobial resistance before and after the ban
      on AGPs, chi-square test was performed.

  9. line 71: Table 1: inconsistency between abbreviation GM and G (table vs. legend). Also, what are the superscript numbers in AMC and SXT? Finally, instead (or additionally) to showing “total 2007-2017” it would be informative to have a column with the difference of the subtotals before minus after ban. That way it would be visible directly which resistances decreased most (and which increased) after vs. before the ban.
    Answer: We revised errata in Table 1. According to reviewer’s opinion, we added column to
    show the difference of subtotals before minus after ban on AGPs.
    Line 94: G → GM
    Supplementary data line 3: G → GM
    Line 100: Table 1 – AMC1, SXT2 → AMC, SXT
    Supplementary data line 1: Table 1 - AMC1, SXT2 → AMC, SXT

  10. line 79: patterns of resistance? Rather “percentage of isolates resistant to 7 subclasses”. In addition, it is more common to define multidrug resistance based on resistance to different classes (or for a more clinically relevant context “categories” as defined by Magiorakos et aline 2012). I could not find a CLSI definition of MDR, if there is one which uses the mentioned subclasses, Table 2 can remain as it is. Otherwise, consider revising.
    Answer: We revised “patterns of resistance” to “percentage of isolates to 7 subclasses” on
    line 73. CLSI divided antimicrobial agents to antimicrobial class including several
      subclasses (eg. Antimicrobial Class-Penicillins including subclasses Penicillin,
      Aminopenicillin, Carboxypenicillin, Amidinopenicillin etc).

        Thank you for your insightful comments. In literal terms, multidrug resistance (MDR)
      means ‘resistant to more than on antimicrobial agent’ but no standardized definitions for
      MDR have been agreed upon yet by the medical community. Many definitions are being
      used in order to characterize patterns of multidrug resistance. Magiorakos AP et al. said
      that the definition most frequently used for bacteria is “resistant to three or more
      antimicrobial subclasses”.So, we think Table can remain as it is.

    Line 73: Before and after the ban on AGPs, the percentage of isolates resistant to 7
      subclasses were 22.9% and 21.0%, respectively, which were the most prevalent.

  11. line 94: better “percentage”, not ratio
    Answer: We revised ratio to percentage on line 21, 73, and 85
    Line 21: ratio → percentage
    Line 73: ratio → percentage
    Line 85: ratio → percentage

  12. line 117: Again, avoid “significantly” when there is no statistical result. Better write “clearly” or similar.
    Answer We revised this sentence on line 119-123.
    Line 119-123: The frequencies of resistance to antimicrobials (gentamicin: 48.4%, neomycin:
    4%, nalidixic acid 70.3%, ampicillin 84.9%, trimethoprim / sulfamethoxazole 60.9%, and
      tetracycline 86.7%) observed in this study are clearly higher than the EU average
      resistance figures (nalidixic acid 59.8%, ampicillin 58.0%, and tetracycline 47.1%), and the
      USA average resistance figures (gentamicin: 23.9%, neomycin 33.8%, ampicillin 68.1%,
      and trimethoprim / sulfamethoxazole 22.0%) [16, 17].

  13. line 119 ff: why compare to Europe (only), when you used CLSI? Is there no US data?
    Answer: We added the data for US and added on line 121.
    Line 121: observed in this study are clearly higher than the EU average.

  14. line 129 ff: this passage is like a repetition of what was already mentioned in line 110 ff. Please omit here or before.
    Answer: We deleted lines 110-115 to avoid repetition.

  15. line 141: were your isolates screened for pathogenicity factors after isolation or was it assumed they are pathogenic because they originated from diarrheic piglets? If it was confirmed please mention it explicitly. Otherwise, pathogenicity is only assumed.
    Answer: We tested virulence factors of isolates in the previous study. For reader friendly, we
    added this information on line 114-116 and line 147
    Line 114-116: In the previous study, we isolated 690 pathogenic coli strains from weaned
      piglets showing signs of enteric colibacillosis from 2007 to 2017, and investigated these
      isolates for adherence (F4, F5, F6, F18, F41, eae, paa, AIDA-I) and toxin (LT, STa, STb,
      Stx2e, EAST-I) genes [4].
    line 147: in this study, we tested the antimicrobial resistance of pathogenic E. coli encoding at
      least one or more virulence factors isolated from pigs with diarrhea

  16. line 155: check phrasing “with  coliwith of US origin”. In addition, did the US study use the same definition of MDR as was used in this study? I also miss a citation here for these 51.7%
    Answer: We added the reference [26] for this phrase. Thank you for your comment. In US
      study, they defined MDR as “resistant to more than on antimicrobial agent“. To make the
      paper reader friendly, we unified the definition of MDR as “resistant to three or more
      antimicrobial subclasses”. Thus, the data 51.7% was changed to 30.9%. This will be find
      on line 168-169.
    Line 168-169: Due to the different types of antimicrobial tested, it is hard to directly compare
      multidrug resistance rates in comparison to the multidrug resistance rates (30.9%) of pigs
      with E. coli of US origin [26], the multidrug resistance rates of Korean piglets were very
      high

  17. line 168: put Enterobacteriaceaein italics
    Answer: We revised this on line 180.

Round 2

Reviewer 1 Report

The authors have answered all of my comments and I am happy to recommend acceptance of the paper.